# Delayed NK Cell Reconstitution and Reduced NK Activity Increased the Risks of CMV Disease in Allogeneic-Hematopoietic Stem Cell Transplantation

**DOI:** 10.3390/ijms21103663

**Published:** 2020-05-22

**Authors:** Ki Hyun Park, Ji Hyeong Ryu, Hyunjoo Bae, Sojeong Yun, Joo Hee Jang, Kyungja Han, Byung Sik Cho, Hee-Je Kim, Hyeyoung Lee, Eun-Jee Oh

**Affiliations:** 1Department of Biomedicine & Health Sciences, Graduate School, The Catholic University of Korea, Seoul 06591, Korea; neojkl1004@naver.com (K.H.P.); jaydcom8673@gmail.com (H.B.); dbsthwjd789@naver.com (S.Y.); eertrqjjhyu4@naver.com (J.H.J.); 2Department of Laboratory Medicine, Seoul St. Mary’s Hospital, College of Medicine, The Catholic University of Korea, Seoul 06591, Korea; hyesungsee@naver.com (J.H.R.); hankja@catholic.ac.kr (K.H.); 3Department of Hematology, Catholic Hematology Hospital, Seoul St. Mary’s Hospital, College of Medicine, The Catholic University of Korea, Seoul 06591, Korea; cbscho@catholic.ac.kr; 4Department of Laboratory Medicine, Catholic Kwandong University International St. Mary’s Hospital, Incheon 22711, Korea; shomermaid@catholic.ac.kr

**Keywords:** cytomegalovirus, hematopoietic stem cell transplantation, acute myeloid leukemia, NK cells, NK cell-mediated antibody-dependent cellular cytotoxicity, NK subsets, IFNγ

## Abstract

Cytomegalovirus (CMV) infection has a significant impact in patients after allogeneic hematopoietic stem cell transplantation (HSCT). We investigated natural killer (NK) cell reconstitution and cytotoxic/cytokine production in controlling CMV infection, especially severe CMV disease in HSCT patients. Fifty-eight patients with acute myeloid leukemia (AML) who received allo-HSCT were included. We monitored NK reconstitution and NK function at baseline, 30, 60, 90, 120, 150, and 180 days after HSCT, and compared the results in recipients stratified on post-HSCT CMV reactivation (*n* = 23), non-reactivation (*n* = 24) versus CMV disease (*n* = 11) groups. The CMV disease group had a significantly delayed recovery of CD56dim NK cells and expansion of FcRγ-CD3ζ+NK cells started post-HSCT 150 days. Sequential results of NK cytotoxicity, NK cell-mediated antibody-dependent cellular cytotoxicity (NK-ADCC), and NK-Interferon-gamma (NK-IFNγ) production for 180 days demonstrated delayed recovery and decreased levels in the CMV disease group compared with the other groups. The results within 1 month after CMV viremia also showed a significant decrease in NK function in the CMV disease group compared to the CMV reactivation group. It suggests that NK cells’ maturation and cytotoxic/IFNγ production contributes to CMV protection, thereby revealing the NK phenotype and functional NK monitoring as a biomarker for CMV risk prediction, especially CMV disease.

## 1. Introduction

Hematopoietic stem cell transplantation (HSCT) is a well-established treatment for patients with acute myeloid leukemia (AML) [1]. While allogeneic HSCT from an human leukocyte antigen (HLA)-matched related donor is ideal, alternative approaches, such as HSCT from HLA-matched unrelated donor (MUD) or haploidentical HSCT from the familial mismatched donor (FMD), are proposed for patients with advanced leukemia [2]. HSCT from MUD and FMD has demonstrated a higher incidence of CMV viremia and delayed immune constitution compared to HSCT from HLA-matched related donors, suggesting the employment of novel approaches, such as immunotherapy or more aggressive CMV prophylaxis, in these patients [3].

Primary CMV infection usually occurs asymptomatically, but virus infection could be a potentially life-threatening complication in immune-deficient patients [4,5,6]. After HSCT, patients are vulnerable to CMV infection, which can occur during immunosuppression and proceed to severe CMV disease with high mortality [4,7,8,9]. The range of CMV infection is wide, ranging from CMV reactivation and presenting mainly as asymptomatic viremia, DNAemia, or antigenemia to CMV end-organ diseases, such as esophagitis, gastroenteritis, hepatitis, retinitis, pneumonia, and encephalitis [5,10]. Currently, preemptive therapy-based monitoring of CMV viremia is performed to improve CMV-related outcomes [5]. The range of CMV infection after HSCT can be different because HSCT patients have diverse CMV serostatus, different CMV prophylaxis, conditioning methods, genetic polymorphism, and inconstant post-HSCT immune reconstitution against CMV infection [5,7,9,11,12]. Considering these differences, CMV-specific immunity plays a vital role in controlling CMV reactivation and disease development. Therefore, verification and monitoring of the patient’s immune cells can be an important tool for the defense of CMV infection.

Natural killer (NK) cells are the initial lymphocyte population and NK cell precursors experience a maturation process during immune reconstitution after HSCT [13,14]. It is well known that NK cells play a crucial role in defense mechanisms against CMV infections before complete recovery of the adaptive immune response, and immune response mediated by NK cells during an early post-HSCT period can contribute towards controlling CMV infection in HSCT patients [13,14,15,16,17]. Previous studies reported that CMV reactivation in HSCT patients promoted NK cell maturation and expansion of NKG2C+ NK cells with specific function against CMV infection [18,19,20]. However, NK reconstitution and associated cytotoxic/cytokine- production in controlling the CMV infection, especially severe CMV disease, is not fully clarified in HSCT patients. 

In the present study, NK immune responses were monitored in AML patients who received HSCT from MUD or FMD by testing the reconstitution kinetics of NK populations and NK-mediated cytotoxic/cytokine production. We compared the results in HSCT recipients stratified on post-HSCT CMV reactivation, non-reactivation versus CMV disease to evaluate the impact of early NK reconstitution in controlling the severity of CMV disease.

## 2. Results

### 2.1. Patient Population

The clinical characteristics of 58 patients according to the CMV reactivation or CMV disease are shown in Table 1. During post-HSCT 180 days, CMV reactivation (CMV viremia > 1000 copies/mL) was detected in 34 (58.6%) patients, and 11 of them were diagnosed with CMV disease, including pneumonitis or gastrointestinal disease. All the patients were divided into three groups as (1) no CMV infection (*n* = 24), (2) CMV reactivation without CMV disease (CMV reactivation) (*n* = 23), and (3) CMV disease group (*n* = 11). The median time to the first detection of CMV viremia was 35 days (range: 15–94 days) and 28 days (range: 19–90 days) after HSCT in CMV reactivation and CMV disease groups, respectively (*p* > 0.05). Of 58 patients, all grades of acute graft-versus-host disease (aGVHD) were experienced by 35 (60.3%) patients and it was identified that aGVHD was not associated with CMV infection or CMV disease (*p* > 0.05). Patients who received reduced-intensity conditioning treatment were identified to be more susceptible to CMV infection (*p* = 0.016).

### 2.2. Reconstitution of Immune Cells against CMV Infection after HSCT

We monitored the absolute number of peripheral white blood cells (WBCs), lymphocytes, CD3+T cells, NK cells, and NKT cells in the three groups. As shown in Figure 1, no difference was observed in the total WBC count. However, the reconstitution of CD3+T cells, NK cells, and NKT cells exhibited a significant decrease in the CMV disease group (Figure 1a) compared to the CMV reactivation or no CMV infection group. Especially, at 60 days post-HSCT, the recoveries of lymphocytes, including CD3+T cell, NK cells, and NKT cells, were significantly reduced in the CMV disease group compared to the CMV reactivation group or no CMV infection group (all *p* < 0.05). 

Subsequently, we examined the recovery of immune cells at 3–4 weeks post-CMV viremia in 34 HSCT patients to investigate whether control of CMV infection induces immune cell redistribution. In patients with progression to CMV disease, a decrease in lymphocyte counts was observed compared to patients with CMV reactivation only (*p* = 0.033) (Figure 1b). Additionally, we generated the ratios between values obtained from the CMV reactivation/CMV disease group and values from the no infection group, and plotted them from before CMV viremia until 150 days after CMV viremia. The ratio values were below one before onset, and the CMV disease group showed delayed recoveries of immunes cells compared to the CMV reactivation group (Figure A3). 

### 2.3. Recovery of NK Subsets against CMV Infection

We distinguished CD56dim and CD56bright NK subsets according to the levels of CD16 and CD56 in the CD3-CD56+NK cells population. Additionally, the levels of FcRγ-CD3ζ+NK subsets (g-NK cells) that are known to be the CMV memory-like NK cells were measured. In a sequential assessment, the recovery of absolute CD56dim, CD56bright, and CMV memory-like FcRγ-CD3ζ+NK cells count differed significantly among the three groups (Figure 2). Median absolute counts of CD56dim, CD56bright, and FcRγ-CD3ζ+NK cells on post-HSCT 60 days were significantly lower in patients with CMV disease (all *p* < 0.05). In addition, the CMV disease group had a significantly delayed recovery of CD56dim NK cells and FcRγ-CD3ζ+NK cells as their expansion started after post-HSCT 150 days. Together, these findings suggest that CMV disease delays the expansion of mature NK cells presented by CD56dim and FcRγ-CD3ζ+NK cells. In the sequential analysis of the ratios (CMV disease or reactivation-to-no infection) during the post-CMV viremia period, the CMV disease group showed delayed reconstitution of NK subsets compared to the CMV reactivation group (Figure A4).

In terms of the distribution of the NK subpopulation, the median percentages of CD56dim cells were higher at all the examined time points. On the contrary, the median percentage of CD56bright cells demonstrated an increase during the first 30–60 days in the CMV reactivation group and remained lower thereafter (<10%) at all times during the 6 months after HSCT, without any significant differences among the three groups. Whereas the median percentage of the FcRγ-CD3ζ+NK population demonstrated an increase after 30 days and remained higher until 180 days after HSCT. However, the level of the FcRγ-CD3ζ+NK percentage after HSCT was not different among the three groups at all time points examined.

### 2.4. Distribution of NK Receptors (NKG2D, NKG2A, NKG2C)-Positive Population in Response to CMV Reactivation or CMV Disease after HSCT

We assumed that the lower recovery and reduced count of NK cells and NK subsets after HSCT is associated with the down- or upregulation of NK cell surface receptor expression. Therefore, we additionally analyzed the expression of several surface molecules (NKG2D, NKG2A, NKG2C, and CD57) present on NK cells, and compared the results among the three groups. The absolute numbers of NKG2D+NK and NKG2A+NK started to increase at 30 days after HSCT in all the three groups; whereas, the absolute number or NKG2D+NK cells at post-HSCT 60 days was significantly decreased in the CMV disease group compared to the CMV reactivation group (*p* = 0.043) (Figure 3). In addition, the absolute count of NKG2D+NK cells recovered to the normal levels in the CMV disease group at post-HSCT 150 days, suggesting a delayed recovery of NKG2D+NK cells. Notably, patients with CMV reactivation or CMV disease displayed expansion of NKG2C+CD57+NK cells at 150–180 days after HSCT, in whom the median percentage of NKG2C+CD57+NK cells were significantly increased from day 30 to 150 after HSCT (patients with CMV reactivation, 2.4% (day 30) vs. 16.1% (day 150), *p* = 0.003; patients with CMV disease, 6.4% (day 30) vs. 34.2% (day 150), *p* = 0.023) (Figure 3b). On the contrary, NKG2C+CD57+NK cells expansion from day 30 to 150 was not observed in HSCT patients without CMV reactivation (3.6% vs. 10.4%, *p* > 0.05). We also confirmed expansion of the NKG2C+CD57+NK cells in the CMV reactivation and CMV disease groups using the ratios (CMV disease or reactivation-to-no infection) (Figure A5).

### 2.5. NK Cytotoxicity against CMV Infection after HSCT

As the CMV disease group showed lower recovery of lymphocytes and delayed expansion of NKG2C+CD57+NK cells after HSCT, we analyzed whether NK function was also decreased in the CMV disease group by testing NK cytotoxicity in response to HLA class I-negative K562 cells (NK-K562 cytotoxicity) and CD20+B lymphoma Raji cells (NK cell-mediated antibody-dependent cellular cytotoxicity; NK-ADCC). 

NK-K562 cytotoxicity was recovered (above 40%) at day 30 after HSCT in all three groups. The recovered NK function was maintained until 120 days in the CMV reactivation group but not in the CMV disease group (Figure 4a). Comparison of the NK cytotoxicity levels (%) within 1 month after the first viremia detection showed adequate NK function (above 40%) in 50.0% of the CMV reactivation group and 33.3% of the CMV disease group, although statistical significance was not found (Figure 4b). 

In terms of NK-ADCC, the HSCT patients without CMV infection and with CMV reactivation only demonstrated satisfactory NK-ADCC results (>30%) at post HSCT 30 days with sustained NK-ADCC levels until post-HSCT 180 days. On the contrary, the patients who developed CMV disease revealed decreased NK-ADCC levels with delayed recovery until post-HSCT 90 days (Figure 4a). Similarly, the NK-ADCC level at less than 1 month after CMV viremia exhibited a significant decrease in patients with CMV disease compared to the patients with CMV reactivation (13.5 [95% CI: 6.4–30.6] vs. %, 42.2 [95% CI: 29.8–57.5], *p* = 0.002) (Figure 4b). When we analyzed the ratios of NK-ADCC results (CMV disease or reactivation-to-no infection), the CMV disease group showed the ratio below one until 120 days after CMV (Figure A6b). We also performed multiple regression analysis to evaluate the relationship between NK-ADCC levels and other factors, including acute GvHD, HSCT intensity, and ATG dose. The NK-ADCC results during the early period of viremia were an independent predictor for CMV disease (*p* = 0.003). In addition, we also plotted the NK-ADCC levels using the ratios (CMV disease or reactivation-to-no infection) in the subgroup regarding acute GvHD, HSCT intensity, or ATG (Figure A7). There were no significant effects of acute GvHD, HSCT intensity, or ATG on the NK-ADCC levels in the CMV infection group. These outcomes suggest that a lower recovery and decreased NK cytotoxic function influences the development of CMV disease in HSCT patients.

### 2.6. NK Function for IFNγ Secretion against CMV Infection after HSCT

Previous studies demonstrated that CMV reactivation induces the expansion of IFNγ, producing NK cells with memory-like features, and these NK cells might contribute towards controlling CMV infection in HSCT patients [16,20]. Therefore, we investigated NK activity for IFNγ secretion to examine CMV infection, especially whether the development of CMV disease is related to lower NK activity in HSCT patients. The reference value of NK-IFNγ levels was defined as above 100 pg/mL.

In HSCT patients without CMV infection, NK-IFNγ levels were markedly increased at 30 days after HSCT and gradually decreased until 120 days. The patients with CMV reactivation also demonstrated increased NK-IFNγ levels at 30 days after HSCT and sustained levels until 90 days after HSCT. Whereas, patients with CMV disease showed markedly decreased NK-IFNγ levels until post-HSCT 120 days (Figure 5a). When we compared NK activity levels for IFNγ secretion at post-HSCT 60 days between the CMV disease and CMV reactivation group, the CMV disease group had significantly decreased levels compared with the CMV reactivation group (9.8 [95% CI: 0.0–384.7] pg/ml vs. 822.4 [95% CI: 129.6–2396.6] pg/mL, *p* = 0.011) (Figure 5a). The results within 1 month after CMV viremia also showed significantly decreased NK-IFNγ levels in the CMV disease group compared to the CMV reactivation group (13.7 [95% CI: 0.0–109.5] pg/mL vs. 489.7 [95% CI: 50.8–1164.9] pg/mL, *p* = 0.016) (Figure 5b and Figure A6).

## 3. Discussion

In the present study, we monitored the sequential data of 58 AML patients who received HSCT from MUD or FMD. Despite the improvement in CMV prophylaxis and the understanding of NK reconstitution in response to CMV infection, few studies have established FMT/MUD in HSCT patients. Additionally, we demonstrated the functional and phenotypic diversities in NK cells in correlation with CMV disease in HSCT patients. The main purpose of this study was to determine whether NK reconstitution and NK cell function are related to controlling early CMV infection after HSCT.

The absolute count of the immune cells during post-HSCT 180 days was monitored to investigate whether the immune reconstitution is related to CMV infection. The total WBC counts remained above the normal reference level until 180 days after HSCT and no difference was observed among the three groups. As per the results of lymphocyte reconstitution after HSCT, CMV reactivation induced an early recovery of lymphocytes, including T, NK, and NKT cells, at 30–60 days. However, the CMV disease group revealed a slower recovery of lymphocytes than in the CMV reactivation or no infection groups. NK cells start reconstituting within the first weeks after HSCT and are known as firstly reconstituted lymphocytes after HSCT [21,22,23]. Previous studies reported that CMV reactivation promoted a rapid NK-cell maturation after allo-HSCT [24,25]. On the contrary, Zhang et al. reported that CMV infection after allo-HSCT did not alter the distribution of peripheral NK cells until 100 days post-HSCT [8]. However, they did not distinguish patients with CMV disease from patients with CMV reactivation. Herein, we observed a delayed recovery of T, NK, and NKT cells in only patients with CMV disease. These findings suggest that delayed reconstitution of T, NK, and NKT cells may increase the risks of CMV disease development in MUD or FMD recipients. 

The initial results of the study indicated a delayed recovery in lymphocyte reconstitution in the CMV disease group compared to the other groups. Furthermore, we analyzed the immune reconstitution of NK subsets (CD56dim, CD56bright, and FcRγ-CD3ζ+NK) and NK receptors (NKG2D, NKG2A, and NKG2C) expressing NK subpopulations. NK cells can be divided into CD56dim NK subsets that cause cytotoxicity and CD56bright NK subsets that are capable of secreting the proinflammatory cytokines [26]. There exist reports on NK subsets specialized for CMV defense, such as the FcRγ-CD3ζ+NK cells and NKG2C+NK cells [27,28,29,30,31]. It has been reported that CMV infection characterizes the NK cell receptor repertoire and effect on the NK cell development and function [14,32,33]. Although the adaptive immunity of NK cells induced by CMV infection may vary among individuals, CMV infection-induced NK cells display clonal expansion, enhanced effector function, and epigenetic modification [34]. In concordance with the previous reports, significantly delayed recovery of CD56dim and FcRγ-CD3ζ+NK cells was observed in the CMV disease group.

The function of NK cells is regulated by diverse surface receptors that transmit either activating or inhibitory signals into NK cells [16,35]. During the patients’ CMV interplay, CMV infection shapes the NK cell receptor repertoire and induces the expansion of an NK cell subset expressing the activating NKG2C receptor. Expansion of the NKG2C+NK cell after HSCT has been reported and the NK cell subset has been proposed to play a role in the resolution of CMV infection [13,14]. Cichocki et al. reported a significant expansion of CD56dimCD57+NKG2C+NK cells in response to CMV reactivation [28]. Adams, *n*. M. et al. reported that the level of NKG2C+NK reconstitution tends to be lower during the initial HSCT days (15–60 days) with a subsequent increase after 200–360 days post-HSCT [15]. In line with these reports, our study also showed an increased percentage of NKG2C+ and CD57+NK cells from day 30 to 150 after HSCT in patients with CMV reactivation and CMV disease. However, the levels of NKG2C+ or CD57+NK cells were not different between the CMV reactivation and CMV disease groups (*p* > 0.05). While percentages provide the frequency of positively expressing NK cells at a population level, mean fluorescence intensity (MFI) reflects the molecules per cell for surface receptors expressed by individual NK cells and can adequately characterize NK cells after CMV infection [14]. Thus, further studies are needed to investigate the MFI value of NK cell surface receptors in association with CMV infection.

CMV reactivation has been reported to promote NK-cell maturation and expansion of CMV-specific NK cells (CMV-induced memory-like NK cells) after HSCT [22,24,36], but studies with an intensive focus on the NK activity and NK cytotoxicity in association with CMV disease after HSCT from MUD or FMD are rare. We analyzed NK function in addition to NK cells reconstitution for 180 days after HSCT in three groups, including no CMV infection, CMV reactivation, and CMV disease. 

Herein, sequential results of K562 cytolysis and NK-ADCC for 180 days demonstrated delayed recovery and decreased levels in the CMV disease patient group compared with the other groups. Especially, NK ADCC within a month after CMV viremia showed significantly reduced results in the CMV disease group compared to the CMV reactivation group. Among the values of the lymphocyte count, NK-ADCC, and NK-IFNγ within one month after CMV viremia that were significantly different between the CMV disease and CMV reactivation groups (*p* < 0.05), only the NK-ADCC results during the early period of viremia were an independent predictor for CMV disease by multiple regression analysis (*p* = 0.003). Zhang et al. demonstrated the response of CD57+NKG2C+FcRγ-CD3ζ+NK cells to CMV viremia and induction of NK-ADCC in HCMV-infected individuals as a mechanism of NK-ADCC against CMV viremia [37]. Similarly, our results also suggest that delayed maturation or NK cells, especially decreased levels of NKG2C+g-NK cells, may induce decreased ADCC in the CMV disease group.

During primary CMV infection, NKG2C+NK cells expand and produce IFNγ. Subsequently, adaptive NK cells (CD56dimCD57+ NKG2C+) increase and become functionally active upon CMV infection [20]. IFNγ secretion from the NK cells is caused by the activation of intracellular signal transduction pathways and stimulation with other cytokines [38,39,40,41]. We finally investigated IFNγ secretion as a marker of NK activity in three groups of patients. The IFNγ secretion using NK Vue-Kit was mainly from NK cells, and T cells or NKT cells play only a minor role [42]. Figure 4 and Figure 5 reveal that cytokine secretion and cytotoxic function do not consistently coexist in reconstituting NK cells after HSCT. In our cohort of patients, the CMV disease group showed delayed expansion of the memory NK population of NKG2C+NK cells along with a delayed display of IFNγ-producing capabilities. However, the levels of NK-IFNγ in CMV reactivation patients remained high for 90 days after HSCT compared to normal patients and CMV disease patients. In concordance with the outcomes of the previous studies, it is proposed that expanded memory-like NKG2C+CD57+NK cells are functionally competent regarding cytokine production and may be involved in the regulation of NK cell’s adaptive immunity [17,43]. However, we cannot define the precise mechanisms affecting the functional regulation of NK cells in CMV infection. Further studies are needed to gain insights on the intracellular signals and the molecular mechanism responsible for the delayed expansion of NKG2C+NKcells and reduced NK function, especially CD107a degranulation in CMV disease patients after HSCT. 

Altogether, our results demonstrate that both NK cells’ maturation and cytotoxic/IFNγ production can contribute to CMV protection, thereby revealing the NK phenotype and functional NK monitoring as biomarkers for CMV risk prediction, especially in CMV disease. As NK cells play a crucial role in mediating early immunity in HSCT, it is hypothesized that the probability to control NK cell reconstitution and function in HSCT recipients may induce important clinical benefits.

## 4. Materials and Methods

### 4.1. Patients

Fifty-eight adult patients who had high-risk AML and underwent allogeneic HSCT at the Seoul St. Mary’s Hospital between August 2015 and November 2018 were enrolled in this prospective cohort study. In the absence of an HLA-matched sibling donor, 33 patients underwent HSCT with HLA matched unrelated (MUD) and 25 patients underwent HSCT with a familial mismatched donor (FMD) with conditioning regimens based on our institution’s transplantation protocol [2]. This study was approved by the Institutional Review Board of Seoul St. Mary’s Hospital (KC12OIST0814). We received written consent from all the patients and collected heparinized whole blood for sequential monitoring of NK reconstitution and NK function tests (0, 30, 60, 90, 120, 150, and 180 days after HSCT). For the CMV prophylaxis, high-dose i.v. acyclovir (10 mg/kg 3 times a day) was administered from the start of conditioning until engraftment (from days −7 to days +28). From the time of neutrophil engraftment to hospital discharge, patients were monitored for CMV infection twice a week with a real-time polymerase chain reaction (PCR)-based assay for CMV DNA using the AccuPower CMV quantitative PCR kit (Bioneer, South Korea). CMV infection was diagnosed according to quantitative real-time PCR (levels (CMV DNA >1000 copies/mL in whole blood). CMV DNA load-guided preemptive therapy and diagnosis of CMV disease were conducted using previously published criteria [44]. 

### 4.2. Immunophenotyping

Peripheral blood mononuclear cells (PBMCs) were separated from fresh blood samples by Ficoll-Hypaque gradients (Sigma-Aldrich, St. Louis, MO, USA) and processed within 4 h. The reconstitution of NK cells after HSCT was determined by eight-color multiparameter flow cytometry using the following mAbs: anti-CD45-FITC (clone; HI30, BD bioscience, San Diego, CA, USA), anti-CD3-V450 (clone; UCHT1, BD bioscience, San Diego, CA, USA), anti-CD16-V500 (clone 3G8; BD bioscience, San Diego, CA, USA), anti-CD56-PE-Cy7 (clone; B159, BD bioscience, San Diego, CA, USA), anti-NKGD-APC (clone; 1D11, BD bioscience, San Diego, CA, USA), anti-NKG2A-PE (clone; REA110, Miltenyi Biotec, Bergisch Gladbach, Germany), anti-NKG2C-Alexa700 (clone 134591, R&D Systems Inc., Minneapolis, MN, USA), and anti-CD57-V450 (clone; TB01, ebioscience, San Diego, CA, USA). PBMCs were also intracellularly stained with anti-CD3ζ-PE (clone; 6B10.2, BD bioscience, San Diego, CA, USA) and anti-FcεRIγ-FITC (FcRγ) (Millipore, Merck Millipore, Burlington, MA, USA) for the analysis of FcεRIγ-deficient NK cells (g-NK cells). NK cell gating and analyses of the NK subpopulation are shown in Figure A1. Flow cytometry data were collected on an FACS Fortessa instrument (BD bioscience, San Jose, CA, USA), using FlowJo version 10.0.6 software (Tree Star, Ashland, OR, USA).

### 4.3. NK Target Cell Lines (K562 and Raji)

The human erythroleukemia cell line K562 was maintained in Dulbecco’s Modified Eagle’s Medium (DMEM, Welgene, Gyeongsan-si, Gyengsangbuk-do, Korea) supplemented with 10% fetal bovine serum, 100 U/mL of penicillin, and 100 μg/mL of streptomycin (both Gibco-BRL, Waltham, MA, USA). Burkitt lymphoma cell line Raji was cultured in RPMI1640 medium containing 10% fetal bovine serum, 100 U/mL of penicillin, and 100 μg/mL of streptomycin (both Gibco-BRL, Waltham, MA, USA) at 37 °C in a 5% CO_2_ incubator.

### 4.4. NK Cytotoxicity

Target cell lines K562 and Raji were labeled with 2 μM CFSE (Thermo Fisher Scientific Inc., New York, NY, USA) to discriminate target cells from effector cells as previously described [45]. Effector peripheral blood mononuclear cells (PBMCs) were incubated with CFSE-labeled target cells at different effecter-to-target (E:T) ratios with 32 and 16 for 4 h at 37 °C, 5% CO_2_. PBMCs were cultured in 150 μL of culture media and 10,000 target cells were used constantly. The negative control with K562 and Raji alone was also incubated in each test. To analyze NK-ADCC, Raji cells were additionally treated with 5 μM Rituximab (MabThera; Roche, Basel, Switzerland) before incubation with effector cells. After incubation, the cell mixture was stained with 10 μL of 7-AAD (Beckman Coulter, Coulter, Fullerton, CA, USA) for 15 min in the dark. Target cells stained with CFSE were analyzed on FACS Fortessa (BD) as shown in Figure A2. NK cytotoxicity (%) was calculated as [cells positive for both CFSE and 7-AAD/total CFSE positive target cells], after subtracting the spontaneous lysis (%) in the negative control. 

### 4.5. NK-IFNγ Secretion Assay

NK-IFNγ secretion assay was performed by enzyme immunoassay using NK Vue-Kit (ATgen, Sungnam, Korea) as described previously [46,47]. Fresh whole blood (1 mL) was attained using tubes containing Promoca. Promoca is a stimulatory cytokine that can specifically stimulate NK cells. The main cell population secreting IFNγ after stimulating whole blood with Promoca was NK cells. After incubation at 37 °C for 20–24 h, cell-free supernatants were harvested and stored at −70 °C until the measurement of IFNγ levels according to the manufacturer’s instructions. Briefly, 50 μL of six standards, controls, and samples were incubated in an antihuman IFNγ-coated plate at room temperature for 2 hours and washed with washing buffer. IFNγ conjugate was added and further incubated at room temperature for 1 hour. After washing and incubation with 100 μL of the substrate at room temperature for 30 min in the dark, the absorbance value was measured at 450 nm. Concentrations of IFNγ were determined with a calibration curve. The measuring range was 0.1–4000 pg/mL and the total imprecision for two levels of controls was less than the 15% coefficient of variations. 

### 4.6. Statistical Analysis

Results are presented as the median with a 95% confidence interval (CI). Result graphs are expressed as the median and connecting lines. The between-group differences of the results were compared by Mann Whitney U tests, Kruskal–Wallis tests, and Chi-square tests. A *p* value of ≤ 0.05 was considered as statistically significant. Multiple regression analysis (enter model) for CMV disease prediction was conducted with interfering variables, including acute GvHD, HSCT intensity, and ATG dose. Statistical analyses were performed using MedCalc statistical software version 17.6 (MedCalc Software bvba, Ostend, Belgium). Reference values of the NK cells and the NK subset were defined based on previously published data [48,49]. 

## Figures and Tables

**Figure 1 ijms-21-03663-f001:**
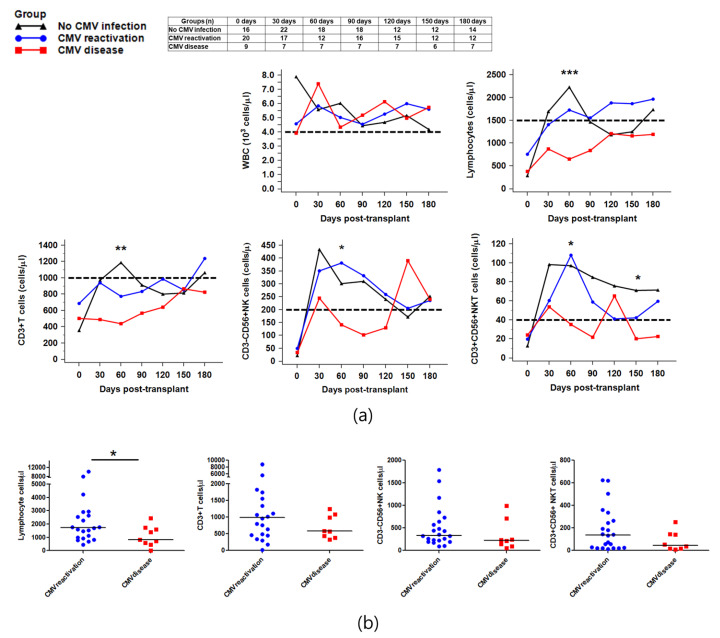
Reconstitution of immune cells in the three cytomegalovirus (CMV) infection groups after hematopoietic stem cell transplantation (HSCT). (**a**) Median absolute count of immune cells on 0, 30, 60, 90, 120, 150, and 180 days after HSCT are demonstrated. The dashed lines indicate reference values of WBC (4000 cells/μL), lymphocytes (1500 cells/μL), CD3+T cells (1000 cells/μL), CD3-CD56+NK cells (200 cells/μL), and CD3+CD56+NKT cell (40 cells/μL). Absolute counts of lymphocyte, T cells, NK cells, and NKT cells at post-HSCT 60 days were significantly decreased in the CMV disease group compared to the CMV reactivation group (*p* < 0.001, *p* = 0.002, *p* = 0.011, and *p* = 0.023, respectively); (**b**) Immune reconstitution within 1 month after CMV viremia detection. The patients with progression to CMV disease showed decreased lymphocytes count compared to patients with CMV reactivation only (*p* = 0.033) (* *p* < 0.05, ** *p* < 0.01, and *** *p* < 0.001).

**Figure 2 ijms-21-03663-f002:**
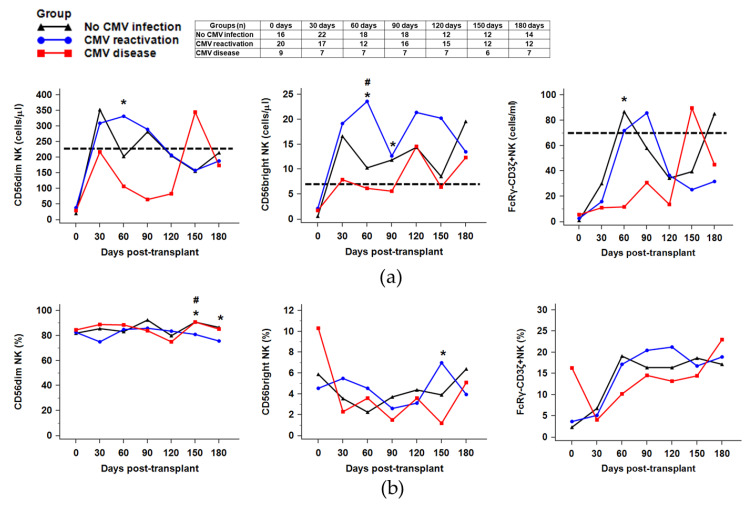
Sequential analysis of NK cell subsets (CD56dim, CD56bright, and FcRγ-CD3ζ+NK subsets) during 180 days after HSCT ((**a**) absolute counts, (**b**) percentage of NK cells). The dashed lines indicate the reference values of CD56dim NK cells (228 cells/μL), CD56bright NK cells (7 cells/μL), and FcRγ-CD3ζ+NK cells (70 cells/μL) on the total NK cell population. The CMV disease group showed delayed recovery and significantly decreased absolute counts of CD56dim, CD56bright, and CMV memory-like FcRγ-CD3ζ+NK cells at post-HSCT 60 days compared to the other groups (*p* = 0.011, *p* = 0.023, and *p* = 0.016, respectively) (* *p* < 0.05). The # symbols indicate significant differences between the CMV reactivation group and no CMV infection group (# *p* < 0.05).

**Figure 3 ijms-21-03663-f003:**
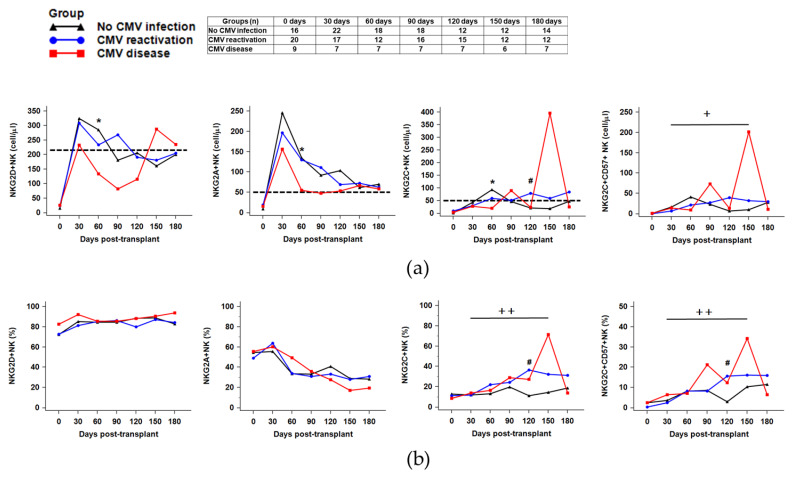
Expression of NK cell receptors (NKG2A, NKG2D, NKG2C, and NKG2C/CD57) at 0, 30, 60, 90, 120, 150, and 180 days after HSCT in the three patient groups ((**a**) absolute counts, (**b**) percentage of receptor expressing NK cells). The dashed lines indicate reference values of NKG2D+NK cells (215 cells/μL), NKG2A+NK cells (50 cells/μL), and NKG2C+NK cells (50 cells/μL) on the total NK cell population. The absolute number of NKG2D+NK cells at post-HSCT 60 days was significantly decreased in the CMV disease group compared to the CMV reactivation group (* *p* = 0.043). The # symbols indicate significant differences between the CMV reactivation group and no CMV infection group (# *p* < 0.05). The percentage of NKG2C+CD57+ NK cells expanded at 150–180 days after HSCT in the CMV reactivation and CMV disease group (CMV reactivation, 2.4% (day 30) vs. 16.1% (day 150), *p* = 0.003; CMV disease, 6.4% (day 30) vs. 34.2% (day 150), *p* = 0.023) (+ *p* < 0.05 and ++ *p* < 0.01).

**Figure 4 ijms-21-03663-f004:**
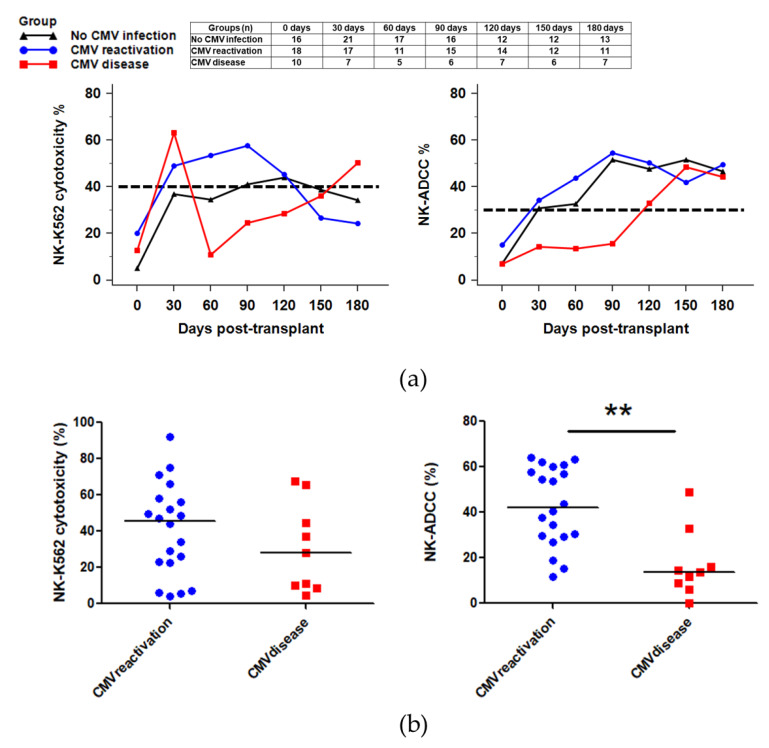
NK-K562 cytotoxicity and NK-ADCC levels in HSCT patients. (**a**) Sequential analysis of NK function until post-HSCT 180 days. The dashed lines indicate the reference values of NK-K562 cytotoxicity (40%) and NK-ADCC (35%). Both NK-K562 cytotoxicity and NK-ADCC levels were less than normal values at post-HSCT 90–120 days in the CMV disease group, suggesting delayed recovery compared to the other groups. (**b**) NK function within 1 month after CMV viremia detection in the CMV reactivation and CMV disease group. NK-ADCC levels at less than 1 month after CMV viremia were significantly decreased in patients with CMV disease compared to the patients with CMV reactivation (13.5 [95% CI: 6.4–30.6] vs. %, 42.2 [95% CI: 29.8–57.5], *p* = 0.002) (** *p* < 0.01).

**Figure 5 ijms-21-03663-f005:**
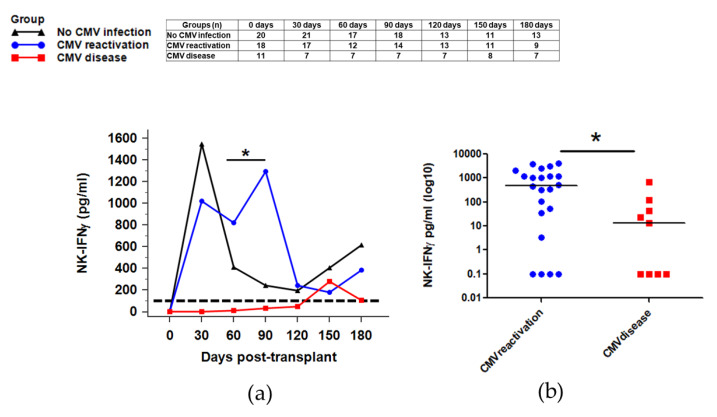
NK function for interferon-gamma (IFNγ) secretion against CMV infection after HSCT. (**a**) The CMV disease group showed markedly decreased NK-IFNγ levels until post-HSCT 120 days, and had significantly decreased levels at 60 days compared with the CMV reactivation group (9.8 [95% CI: 0.0–384.7] pg/mL vs. 822.4 [95% CI: 129.6–2396.6] pg/mL, *p* = 0.011). (**b**) NK-IFNγ levels at less than 1 month after CMV viremia were significantly decreased in the CMV disease group compared to the CMV reactivation group (13.7 [95% CI: 0.0–109.5] pg/mL vs. 489.7 [95% CI: 50.8–1164.9] pg/mL, *p* = 0.016) (* *p* < 0.05). The dashed line indicates the reference value of NK-IFNγ (100 pg/mL).

**Table 1 ijms-21-03663-t001:** Demographics of the studied patients (*n* = 58)**.**

Characteristics	All Participants (*n* = 58)	No CMV Infection (*n* = 24)	CMV Reactivation (*n* = 23)	CMV Disease (*n* = 11)
**Median age (range)**	48 (18–69)	45 (18–61)	48 (32–69)	55 (22–65)
**Sex**; **male**, ***n* (%)**	34 (58.6)	15 (62.5)	14 (60.9)	5 (45.5)
**Donor type**, ***n* (%)**				
MUD	33 (56.9)	17 (70.8)	12 (52.2)	4 (36.4)
FMD	25 (43.1)	7 (29.2)	11 (47.8)	7 (63.6)
**HSCT intensity**, ***n* (%)**				
MAC	29 (50.0)	**17 (70.8) ***	8 (34.8)	4 (36.4)
Non-MAC	29 (50.0)	7 (29.2)	15 (65.2)	7 (63.6)
**GvHD**, ***n* (%)**				
Acute GvHD	35 (60.3)	13 (54.2)	13 (56.5)	9 (81.8)
Chronic GvHD	20 (34.5)	8 (33.3)	7 (30.4)	5 (45.5)
**GvHD prophylaxis**, ***n* (%)**				
Tacrolimus	57 (98.3)	23 (95.8)	23 (100.0)	11 (100.0)
Cyclosporine	1 (1.7)	1 (4.2)	0 (0.0)	0 (0.0)
**ATG dose**, ***n* (%)**				
1.25–2.5 mg	37 (63.8)	17 (70.8)	14 (60.9)	6 (54.5)
5 mg	21 (36.2)	7 (29.2)	9 (39.1)	5 (45.5)
**CMV recipient/donor serostatus**, ***n* (%)**				
R+/D+	55 (94.8)	22 (91.7)	22 (95.7)	11 (100.0)
R+/D−	2 (3.4)	1 (4.2)	1 (4.3)	0 (0.0)
R−/D−	1 (1.7)	1 (4.2)	0 (0.0)	0 (0.0)
**Status**, ***n* (%) during the first 6 Mo**				
Relapse	3 (5.3)	2 (8.3)	1 (4.5)	0 (0.0)
**Time to CMV viremia detection, median day (range)**	34 (15–94)	0 (0–0)	35 (15–94)	28 (19–90)

Abbreviation: MUD, matched unrelated donor; FMD, familial mismatched donor; HSCT, hematopoietic stem cell transplantation; MAC, myeloablative conditioning; GvHD, graft versus host disease; ATG, anti thymocyte globulin; CMV, cytomegalovirus; AML, acute myeloid leukemia. The chi-squared test was used with *p*-value (* *p* < 0.05, No CMV infection group vs. CMV reactivation group).

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
