# Peer review of "Delayed NK Cell Reconstitution and Reduced NK Activity Increased the Risks of CMV Disease in Allogeneic-Hematopoietic Stem Cell Transplantation"

_ijms, 2020, doi:10.3390/ijms21103663_

Round 1

Reviewer 1 Report

The manuscript entitled "Delayed NK cell Reconstitution and Reduced NK Activity Increased the Risks of CMV Disease in Allogeneic-hematopoietic Stem Cell Transplantation " is extremely interesting but additional evaluation should be performed to state such a conclusion. The main problem of the manuscript is that that the immune reconstitution of immune cells and NK cell subsets significantly differ on day 60 after HSCT during the course after HSCT. However, the onset of viremia ranged between 15 and 95 in CMV reactivation and between 19-90 days in disease group resulting in an extremely fluctuation during their course after HSCT. To conclude that a delayed reconstitution happens, the two courses should then be analysed by two-way-ANOVA (CMV and course of values). Another possibility to demonstrate that a delayed reconstitution is associated to CMV, is the generation of ratios i) between the values obtained from CMV reactivation and no CMV infection and ii) between the values obtained from CMV disease and no CMV infection. In case of a delayed reconstitution the values should be below one in the early times points after HSCT. Additionally, the values shoud be stratified before and after onset of CMV reactivation and disease. To concluded that a delayed reconstitution increases the risk of CMV reactivation or disease the values should be below one at time points before onset.

Author Response

The manuscript entitled "Delayed NK cell Reconstitution and Reduced NK Activity Increased the Risks of CMV Disease in Allogeneic-hematopoietic Stem Cell Transplantation" is extremely interesting but additional evaluation should be performed to state such a conclusion. The main problem of the manuscript is that that the immune reconstitution of immune cells and NK cell subsets significantly differ on day 60 after HSCT during the course after HSCT. However, the onset of viremia ranged between 15 and 95 in CMV reactivation and between 19-90 days in disease group resulting in an extremely fluctuation during their course after HSCT. To conclude that a delayed reconstitution happens, the two courses should then be analysed by two-way-ANOVA (CMV and course of values). Another possibility to demonstrate that a delayed reconstitution is associated to CMV, is the generation of ratios i) between the values obtained from CMV reactivation and no CMV infection and ii) between the values obtained from CMV disease and no CMV infection. In case of a delayed reconstitution the values should be below one in the early times points after HSCT. Additionally, the values should be stratified before and after onset of CMV reactivation and disease. To concluded that a delayed reconstitution increases the risk of CMV reactivation or disease the values should be below one at time points before onset.

Answer and correction

Thank you for careful review. We totally agree with reviewer’s opinion. According to the reviewer’s recommendation, we performed Two-way ANOVA (parametric statistics analysis). However, our data are not normally distributed, and Two-way-ANOVA analysis cannot be used.

According to the reviewer’s the other recommendation, we generated the ratios between the values obtained from CMV reactivation/CMV disease group and no infection group. And then, we plotted and compared NK reconstitution and NK function before and after CMV viremia during 150 days (Supplemental Figure A3-A6).

In addition, we added results about the ratios as “Additionally, we generated the ratios between values obtained from CMV reactivation /CMV disease group and values from no infection group, and plotted them from before CMV viremia until 150 days after CMV viremia. The ratio values were below one before onset, and CMV disease group showed delayed recoveries of immunes cells compared to the CMV reactivation group (supplementary Figure A3).” in lines 106-110.

“In the sequential analysis of the ratios (CMV disease or reactivation-to-no infection) during post-CMV viremia period, CMV disease group showed delayed reconstitution of NK subsets compared to CMV reactivation group (supplementary Figure A4).” in lines 132-135.

 “We also confirmed expansion of the NKG2C+CD57+NK cells in CMV reactivation and CMV disease groups using the ratios (CMV disease or reactivation-to-no infection) (supplementary Figure A5).” In lines 169-171.

“When we analyzed the ratios of NK-ADCC results (CMV disease or reactivation-to-no infection), CMV disease group showed the ratio below one until 120 days after CMV (Supplementary Figure A6b).” in lines 202-204.

I hope the revised manuscript will better meet the requirements of the ‘International Journal of Molecular Sciences’ for publication. I thank you again for the constructive review by the referee.

Reviewer 2 Report

In the manuscript submitted by Ki Hyung Park et al., the authors investigated the impact of CMV infection on immune cell reconstitution after HLA mismatched HSCT from 58 AML patients. Although the design of this study is well done, I have some concerns about the analysis and the functional tests used by the authors.

I have the following concerns that can help the authors to consolidate their results:

  1. In CMV disease group (n=11), 9 patients developped acute GvHD, 7 received non-MAC treatment and 6 patients received 1.25-2.5 mg ATG although 5 patients received 5 mg. The authors should evaluate the GvHD, ATG concentration and HSCT intensity impacts on immune cell reconstitution on all patients and particularly in patients with CMV viremia to confirm that these graft parameters (GvHD, ATG concentration and HSCT intensity) do not impact immune lymphocyte reconstitution. This point appears essential and can easily be achieved.
  2.  
  3. NKG2D is expressed on all NK cells as shown in figure S1Ab. Thus the investigation of Mean Fluorescent Intensity of NKG2D on NK cells should be more stringent to evaluate the down or up regulation of NK cell surface receptor expression as proposed by the authors (page 5 line 143).
  4.  
  5. The phenotypic analysis was performed by multicolor flow cytometry and it is shame that NK cell functions as degranulation (CD107a) or IFNg production were not investigated in the same way by flow cytometry. Indeed, NK-IFNg secretion assay using NK Vue-Kit and NK cytotoxicity using CFSE are less convincing than standard approaches. I make a reservation about the NK Vue Kit as the comparison of this approach with the standard tests has not been demonstrated. As the cytokine (Promoca®) is not identified in NK Vue Kit, a comparison with standard stimulation with HLA class I deficient cells as K562 ou 721.221 cell lines should be performed. Morever, Promoca® triggered all whole blood components as T, NKT and NK cells to produce IFNg. The authors cannot exclude T lymphocytes as IFNg producers. These data should be confirmed by standard assays validated in numerous published studies on NK cells.

I have minor concerns:

  1. In the abstract, “g-NK” is not informative and should be modified to clarify this cell population.
  2.  
  3. The number of studied patients at all time points should be indicated in figure legends (Figure 1a, 2, 3, 4a and 5a).
  4.  
  5. The authors should clarify the notion of reference values used in all figures.

Author Response

Reviewer 2.

  1. In CMV disease group (n=11), 9 patients developped acute GvHD, 7 received non-MAC treatment and 6 patients received 1.25-2.5 mg ATG although 5 patients received 5 mg. The authors should evaluate the GvHD, ATG concentration and HSCT intensity impacts on immune cell reconstitution on all patients and particularly in patients with CMV viremia to confirm that these graft parameters (GvHD, ATG concentration and HSCT intensity) do not impact immune lymphocyte reconstitution. This point appears essential and can easily be achieved.

Answer and correction

Thank you for careful review. We agree with reviewer’s opinion. According to the reviewer`s comment, we performed ‘multiple regression analysis’ to evaluate the relationship between immune cell recovery and other factors including acute GvHD, HSCT intensity and ATG dose. Among the values of lymphocyte count, NK-ADCC and NK-IFNγ within one month after CMV viremia that were significantly different between CMV disease and CMV reactivation groups (P < 0.05), only NK-ADCC results during the early period of viremia was independent predictor for CMV disease (P = 0.003).

We clarified that in result and discussion as ” We also performed multiple regression analysis to evaluate the relationship between NK-ADCC levels and other factors including acute GvHD, HSCT intensity and ATG dose. The NK-ADCC results during the early period of viremia was independent predictor for CMV disease (P = 0.003). In addition, we also plotted NK-ADCC levels using the ratios (CMV disease or reactivation-to-no infection) in subgroups regarding acute GvHD, HSCT intensity or ATG (Supplementary Figure A7). There were no significant effects of acute GvHD, HSCT intensity or ATG on the NK-ADCC levels in CMV infection group. We performed multiple regression analysis to evaluate the relationship between NK-ADCC levels and other factors including acute GvHD, HSCT intensity and ATG dose. The NK-ADCC results during the early period of viremia was independent predictor for CMV disease (P = 0.003).” in lines 204-211.

 “Among the values of lymphocyte count, NK-ADCC and NK-IFNγ within one month after CMV viremia that were significantly different between CMV disease and CMV reactivation groups (P < 0.05), only NK-ADCC results during the early period of viremia was independent predictor for CMV disease by multiple regression analysis (P = 0.003).” in lines 306-310.

“Multiple regression analysis (enter model) for CMV disease prediction was conducted with interfering variables including acute GvHD, HSCT intensity and ATG dose.” in lines 406-407.

  1. NKG2D is expressed on all NK cells as shown in figure S1Ab. Thus the investigation of Mean Fluorescent Intensity of NKG2D on NK cells should be more stringent to evaluate the down or up regulation of NK cell surface receptor expression as proposed by the authors (page 5 line 143).

Answer and correction

We agree with reviewer`s opinion. The measurement of MFI values for NK subsets would be useful markers for immune reconstitution. However, the purpose of present study was to compare the recovery of NK subpopulation and NK function among CMV disease, CMV reactivation and no infection group. Unfortunately, we performed flowcytometric analysis using the absolute count and percentage per lymphocytes rather than MFI expression. When we re-analyze the MFI values in each group according to the reviewer’s recommendation, we could not get valuable data regarding the CMV infection.

According to the reviewer’s comment, we added the limitation of this study in discussion section as “While percentages provide the frequency of positively expressing NK cells at a population level, MFI reflect the molecules per cell for surface receptors expressed by individual NK cell and can adequately characterize NK cells after CMV infection [14]. Thus, further studies are needed to investigate the MFI value of NK cell surface receptor in association with CMV infection.” in lines 293-297.

In addition, we changed the description of “expression of ..” into “ Distribution of NK receptors (NKG2D, NKG2A, NKG2C)-positive population” to clarify the results in lines 153.

  1. The phenotypic analysis was performed by multicolor flow cytometry and it is shame that NK cell functions as degranulation (CD107a) or IFNg production were not investigated in the same way by flow cytometry. Indeed, NK-IFNg secretion assay using NK Vue-Kit and NK cytotoxicity using CFSE are less convincing than standard approaches. I make a reservation about the NK Vue Kit as the comparison of this approach with the standard tests has not been demonstrated. As the cytokine (Promoca®) is not identified in NK Vue Kit, a comparison with standard stimulation with HLA class I deficient cells as K562 ou 721.221 cell lines should be performed. Morever, Promoca® triggered all whole blood components as T, NKT and NK cells to produce IFNg. The authors cannot exclude T lymphocytes as IFNg producers. These data should be confirmed by standard assays validated in numerous published studies on NK cells.

Answer and correction

We agree with reviewer`s opinion. However, in present study, we did not perform the CD107a degranulation assay. In previous study (Park KH et al., BioMed research international. 2013;2013:210726.), we compared CD107a expression with NK cytotoxicity, and reported that there was a weak correlation between CD107a expression and NK cell cytotoxicity by CFSE/7-AAD staining (r=0.3, P<0.001). CD107a expression on NK cells may not necessarily correlate to NK activity, because NK cell cytotoxicity is a stepwise combined process including adhesion, activation and secretion of lytic granules. According to the reviewer`s comment, we added suggestion about the CD107a study in discussion as “Further studies are needed to gain insights on intracellular signals and the molecular mechanism responsible for the delayed expansion of NKG2C+ cells and reduced NK function, especially CD107a degranulation in CMV disease-patients after HSCT. ” in lines 329-332.

Promoca is proprietary NK stimulating cytokine cocktail. However, the exact composition of Promoca is not informed by the manufacturer. Nederby L. et al., performed intracellular flow cytometry and proved that the predominant source of IFNγ in NK this assay was NK cells (J. Immunol. Methods 2018, 458, 21–25). Minor faction of secreted IFNγ was from T cell and NKT cells, but they concluded the secreted IFNγ after incubated with Promoca was reflection of NK cell activity in whole blood. We clarify that as “The IFNγ secretion using NK Vue-Kit was mainly from NK cells, and T cells or NKT cells play only a minor role [42].” in lines 319-320.

I have minor concerns:

1 .In the abstract, “g-NK” is not informative and should be modified to clarify this cell population.

  1. The number of studied patients at all time points should be indicated in figure legends.
  2. The authors should clarify the notion of reference values used in all figures.

Answer and correction

Thank you for your detailed comments.

  1. We changed “g-NK” into “FcRγ-CD3ζ+NK” in the entire manuscript
  2. We included the number of studied patients in all figures.
  3. We included the reference values in each figure legend.

I hope the revised manuscript will better meet the requirements of the ‘International Journal of Molecular Sciences’ for publication. I thank you again for the constructive review by the referee.

Round 2

Reviewer 1 Report

No further comments.

Reviewer 2 Report

The authors have improved their manuscript taking into account all reviewer comments.